# PD-L1, CD4+, and CD8+ Tumor-Infiltrating Lymphocytes (TILs) Expression Profiles in Melanoma Tumor Microenvironment Cells

**DOI:** 10.3390/jpm13020221

**Published:** 2023-01-27

**Authors:** Bogdan Marian Caraban, Elena Matei, Georgeta Camelia Cozaru, Mariana Aşchie, Mariana Deacu, Manuela Enciu, Gabriela Izabela Bălţătescu, Anca Chisoi, Nicolae Dobrin, Lucian Petcu, Emma Gheorghe, Laurențiu-Tony Hangan, Mihai Cătălin Roșu, Cristian Ionuț Orasanu, Antonela-Anca Nicolau

**Affiliations:** 1Faculty of Medicine, “Ovidius” University of Constanta, 900470 Constanta, Romania; 2Center for Research and Development of the Morphological and Genetic Studies of Malignant Pathology, “Ovidius” University of Constanta, 900591 Constanta, Romania; 3Clinical Service of Pathology, “Sf. Apostol Andrei” Emergency County Hospital, 900591 Constanta, Romania; 4Dental Medicine Faculty, “Ovidius” University of Constanta, 900470 Constanta, Romania

**Keywords:** PD-L1, CD4+, CD8+, tumor-infiltrating lymphocytes, melanoma tumor microenvironment cells

## Abstract

(1) Background: Because melanoma is an aggressive tumor with an unfavorable prognosis, we aimed to characterize the PD-L1 expression in melanomas in association with T cell infiltrates because PD-1/PD-L1 blockade represents the target in treating melanoma strategy. (2) Methods: The immunohistochemical manual quantitative methods of PD-L1, CD4, and CD8 TILs were performed in melanoma tumor microenvironment cells. (3) Results: Most of the PD-L1 positive, expressing tumors, have a moderate score of CD4+ TILs and CD8+TILs (5−50% of tumor area) in tumoral melanoma environment cells. The PD-L1 expression in TILs was correlated with different degrees of lymphocytic infiltration described by the Clark system (X^2^ = 8.383, *p* = 0.020). PD-L1 expression was observed often in melanoma cases, with more than 2−4 mm of Breslow tumor thickness being the associated parameters (X^2^ = 9.933, *p* = 0.014). (4) Conclusions: PD-L1 expression represents a predictive biomarker with very good accuracy for discriminating the presence or absence of malign tumoral melanoma cells. PD-L1 expression was an independent predictor of good prognosis in patients with melanomas.

## 1. Introduction

Melanoma is a highly aggressive tumor, most often with a cutaneous presentation with an immunogenic character, unfavorable prognosis, and raised mortality rate. Older fair-skinned males represent the high-risk population with a personal or family history of melanoma and with chronic UV exposure [1,2]. In recent years, significant progress in targeted therapy development [3,4] and novel prognostic biomarkers have been necessary for patient treatment strategy tailoring [5]. 

It is recognized that the immune inhibitory signaling pathways have an important role in immunosuppressive microenvironment maintenance, which favor cancer development [6]. An important co-inhibitory pathway is the programmed death-ligand 1 (PD-L1) and programmed death-1 (PD-1) axis [7]. The expression of PD-1 is induced on effector T-cells in response to inflammatory signals, and PD-L1 (B7-H1 or CD274) was the first identified ligand of PD-1 and is expressed in lymphocytes, vascular endothelium, mesenchymal stem cells, neuronal cells, and tumor cells [8]. PD-1/PD-L1 interactions have roles in T-cell-mediated immune responses inhibiting, cytokine production limiting, and tumor immune escape promotion [5,9]. PD-L1 binding to PD-1 suppresses the immune response through CD8+ T cell inhibition and CD4+T-regulatory lymphocyte activation, and melanoma tumor cells use this mechanism [10,11]. PD-L1 is expressed in the tumor microenvironment cells facilitating immune evasion, a predictive biomarker for malignant melanoma evolution [12,13,14]. PD-L1 expression in tumor cells is the most studied predictive biomarker, but a correlation with a therapeutic response is insufficient for melanoma for clinical use. PD-L1 overexpression is examined as a prognostic factor in diverse cancers, such as lung cancer [15], gastric cancer [16], ovarian cancer [17], breast cancer [18], prostate cancer [19], bladder cancer [20], cervical cancer [21], colorectal cancer [22], pancreatic cancer [23], and renal cell carcinoma [24]. Various studies reported the prognostic value of PD-L1 expression in melanoma patients [5,25,26,27,28,29,30,31,32,33,34,35,36,37]. 

Other predictive biomarkers studied in melanoma tumors include T cell receptors and T cell infiltrates. CD8+ T cell presence in the melanoma periphery is associated with a better response to PD-1 inhibitors [38,39]. Most malignant melanocytic tumors evolved despite immune cell presence in the tumor microenvironment, suggesting that these lymphocytes fail to control tumor growth [7,40,41]. The tumor-infiltrating lymphocytes (TILs) were a part of many studies in the past few years [42], bringing important evidence that a high number of lymphocytes are present in the development site of a tumor, with numerous CD8-positive T lymphocytes. These lymphocytes present in melanoma must be differentiated from lymphocytes in a skin lymphoma that has immunoreactivity for CD20 but a negative reaction for CD30 and CD45 [43]. A statistically significant correlation between skin melanoma survivors and increased risk of Non-Hodgkin Lymphoma was observed in the first three years after the diagnosis of skin melanoma [44].

In this study, we used a melanoma patient cohort to characterize the PD-L1 expression in melanoma cases associated with T-cell infiltrates, with roles in diagnosis and prognosis. 

## 2. Materials and Methods

### 2.1. Patient Group

This study collected the cases from the electronic database of the Pathology Department of ”St. Andrew’s” Clinical Emergency County Hospital of Constanta diagnosed with different forms of melanoma in the last two years. We identified 61 patients with surgically removed skin melanocytic tumors upon dermatological recommendation that proved to be melanomas before any oncological treatment. The histologic diagnosis for each specimen was confirmed on Hematoxylin Eosin (HE) stain slides by two experienced pathologists, with a concordance of measurements between 80−90%. We kept a representative tissue block to use for additional studies. Clinico-morphologic features were taken from the pathological reports, including age, gender, Clark level, Brelow thickness, TILs, and vascular and perineural invasion.

Our study followed these criteria: (1) inclusion of patients diagnosed with histologically confirmed melanoma; (2) detection of PD-L1, CD4, and CD8 TILs immunohistochemical expression in the melanomas (IHC) analysis; (3) identification of predictive biomarkers as PD-L1, CD4, CD8 tumor-infiltrating lymphocytes expression patterns; and calculation of the hazard ratio (HR) and 95% confidence interval (95% CI).

### 2.2. TILs Evaluation

TILs evaluation was performed on Hematoxylin Eosin (HE) stain using the Clark scoring system, classifying immune infiltrate as brisk (TILs present throughout the vertical growth), non-brisk (TILs in one or more foci of the vertical growth phase), or absent. 

### 2.3. Immunohistochemical (IHC) Analysis 

The immunohistochemical evaluation was performed with a panel of three antibodies from Master Diagnostica (Granada, Spain) (ready-to-use): PD-L1 (CAL10-clone, rabbit monoclonal antibody), CD4 (EP204-clone, rabbit monoclonal antibody), CD8 (SP16-clone, rabbit monoclonal antibody). We used formalin-fixed paraffin-embedded blocks to obtain four µm-thick sections, 3, 3′diaminobenzidine (DAB) as chromogen, with brown staining. The sections were finally counterstained with Mayer‘s Hematoxylin and mounted. For positive and negative quality controls, we used tonsil tissue according to the manufacturer’s recommendation. The three immunohistochemical biomarkers were evaluated using optic microscopy, considering the positive reaction ≥5%. The degrees of intensities were considered: weakly positive (1+), moderate positive (2+), and intense positive (3+). We assessed the PD-L1 expression in melanoma cells, and TIL’s and CD4 and CD8 TIL’s expression in melanoma tumor microenvironment cells.

### 2.4. Statistical Analysis

The experimental data were analyzed using IBM SPSS Statistics Version 26 (Armonk, NY: IBM Corp). The procedures used were descriptive statistics (for characterization of the discreet and continuous variables defined in the database), parametric test (Independent Sample t Test with Levene’s Test and One Way ANOVA with posthoc Tukey’s range test and Games-Howell Test), Nonparametric tests (Pearson Chi-Square with Cramer’s V test for the association between categorical variables, Mann-Whitney U Test and Kruskal-Wallis H Test for continuous variables). All *p*-values below 0.05 were considered statistically significant. Receiver operating characteristic (ROC) and area under the curve (AUC) made by MedCalc v20.111 Software Ltd. (Ostend, Belgium) were used to establish the accuracy of the biomarkers in melanoma diagnoses. The sensitivity and specificity of biomarkers are represented by the Youden index, which is the optimal cut-off point as the value that maximizes the area under the ROC curve [45,46]. Furthermore, multivariate Cox-proportional hazard regression was performed to determine the potential prognostic values of PD-L1, CD4, and CD8 TILs in melanoma patients.

## 3. Results

### 3.1. Clinico-Pathological Analysis of Melanoma Tumor Microenvironment Cells, Associated with IHC Patterns of PD-L1 (+/-) Expression

PD-L1 (+/-) expression patterns in patients with acral, cervical-cranial, and limb localization of melanomas are summarized in Table 1. The incidence of PDL-1 positive expressions for patients with acral and cervico-cranial localization of melanoma was notably lower than for patients with upper body and limbs localizations of melanoma (TILs-A:5.41%; CC:8.11% vs. UB:51.35%; L:35.14%, *p* < 0.05; MTCs-A:7.10%; CC:17.86% vs. UB:53.62%; L:21.43%, *p* ≥ 0.05). We observed PD-L1 positive expression patterns in 56.76% of TILs and 46.43% in MTCs (*p* ≥ 0.05) for IV grade of Clark level of invasion in melanoma samples from 61 patients when a TILs threshold of positive reaction ≥5%.

When comparing the Breslow tumor thickness between TILs with MTCs, were observed the lowest proportion of PD-L1 (+) expression, at 5.41% (TILs) and 0% (MTCs) for <1 mm diameter, while the highest proportion of PD-L1 (+) specimens was seen for >4mm diameter, at 54.05% (TILs, *p* ≥ 0.05) and 57.14% (MTCs, *p* < 0.05) from cases.

The association between PD-L1 expression and clinicopathological features was analyzed, age (≥65 vs. <65 years), gender (male vs. female), localization (acral, upper body, cervico-cranial, limb), Clark level of invasion (I, II, III, IV, V), Breslow tumor thickness (<1 mm; 1–2 mm; 2–4 mm; >4 mm), lymphovascular invasion (present, absent), perineural invasion (present, absent) (Table 1). 

PD-L1 expression was significantly correlated with melanoma localizations in infiltrating immune cells (X^2^ = 9.603, *p* = 0.018) and Breslow tumor thickness in melanoma tumor cells (X^2^ = 9.933, *p* = 0.014), but no significant relationship between PD-L1 expression and age, gender, or morphological characteristics was seen (Clark level of invasion, lymphovascular invasion, and perineural invasion).

### 3.2. PD-L1 Expression in Tumor Cells and Immune-Infiltrating Cells in Various Morphological Types of Melanoma and Tils Quantification and Characterization

PD-L1 (+/-) expression patterns were observed in the different morphological types of melanomas as lentigo maligna melanoma (LMM), superficial spreading melanoma (SSM), superficial spreading melanoma with vertical growth nodule (SSM-VGN), and nodular melanoma (NM), being presented in Table 2. 

The highest incidences of PD-L1 (+) in TILs were observed for SSM (37.84%) and SSM-VGN (37.84%, *p* ≥ 0.05). Superficial spreading melanoma has in MTCs the highest incidence (46.43%, *p* ≥ 0.05). PD-L1 (+/-) expressions in TILs were observed in strong association with CD8+ and CD4+ T-lymphocytes (X^2^ = 11.238, *p* = 0.003; X^2^ = 12.614, *p* = 0.001, Table 2).

PD-L1 (+) patterns associated with moderate to severe CD8+ and CD4+ lymphocytes intensities (CD8+TILs: 69.57%; MTCs: 75.00%; CD4+TILs: 97.30%; MTCs: 85.71%) compared to mild (CD8+TILs: 2.79%; MTCs: 7.14%, CD4+TILs: 0.00%; MTCs: 3.57%, *p* < 0.05) were observed in the majority of morphological types of melanomas (Table 2) (Figure 1, Figure 2, Figure 3, Figure 4 and Figure 5). Melanomas negative for PD-L1 show a significant degree of staining (2+) by CD4 and CD8 infiltrating lymphocytes (CD8+TILs: 58.53%; MTCs: 75.76%, CD4+TILs: 70.83%; MTCs: 87.88%, *p* < 0.05).

Although PD-L1 expression associated with no TILs was not observed in melanoma tumor cells (X^2^ = 2.775, *p* = 0.514), PD-L1 was expressed in T lymphocytes (X^2^ = 8.383, *p* = 0.020). In our study, PD-L1 showed intertumoral heterogeneity. PD-L1, CD4 TILs, and CD8TILs biomarkers intensities were observed in Figure 1, Figure 2, Figure 3, Figure 4 and Figure 5. 

### 3.3. Predictive and Prognostic Roles of PD-L1, CD4, and CD8 Tumor-Infiltrating Lymphocytes Biomarkers in Melanoma Tumor Microenvironment Cells

To establish the laboratory diagnostic, the predictive model (ROC curves) is used to estimate the risk of patient adverse outcomes in medical research and to use as a discriminatory tool for true positive values (sensitivity) and true negative values (specificity) [45]. ROC curves were used to show the accuracy of the methods by interpreting the true positive results (TPR, sensitivity) and false positive results (FPR, 100-specificity) for each biomarker with two overlapping distributions (negative, positive) (Table 3, Figure 6). Sensitivities of methods in patients with melanoma were very good for PD-L1, CD8TILs, and CD4TILs biomarkers (TILs: 97.30%; M: 96.43%, 89.13%, and 84.31%, *p* < 0.001 (Table 3, Figure 6). Also, specificities of analyzed biomarkers in melanoma tumor microenvironment cells were very good for PD-L1 (TILs: 91.67%; M: 96.97%), CD8 (86.67%), CD4 (90.00%, *p* < 0.001) (Table 3, Figure 6). The area under the curve (AUC) for our studied biomarkers (PD-L1, CD4 TILs, and CD8 TILs) starts from 0.941 to 0.975, which means a very good ability of the ROC test to discriminate the presence or absence of malign tumoral melanoma cells.

Furthermore, multivariate analysis Cox-proportional hazard regression analyzed the following parameters as the PD-L1, CD4, and CD8 TILs expression patterns to observe their potential prognostic roles in melanoma tumor microenvironment cells (Table 4). In our analysis, we observed that PDL-1 represents a favorable prognostic biomarker, an independent predictor factor implied in the prognosis of the melanoma patients. 

## 4. Discussion

Immunotherapy has proved in the past few years to have a real success for many types of cancer. So far, the most effective immunotherapies are the monoclonal antibodies targeting the checkpoint molecules CTLA-4 (cytotoxic T lymphocytes-associated protein 4) and PD-1 (programmed cell death protein 1) and its ligand PD-L1. Antibody blockade of PD-1 and its ligand PD-L1 have antitumor efficacy and durability of responses in melanoma patients [14]. PD-L1 expression by melanoma tumor cells is heterogeneous and contiguous to tumor-infiltrating lymphocyte areas [11]. Melanomas may express PD-L1 against antitumor immune effector cells, facilitating immune evasion even if the B7-H1 costimulatory molecule is present on the surface of tumor cells. This regulates the cellular and humoral immune responses through the PD-1 receptor on activated T and B cells. Also, it was observed that in vitro, the B7-H1 tumor cell lines might increase the apoptosis of antigen-specific human T-cells, which means the apoptotic effects of B7-H1 are mediated by one or more receptors other than PD-1 [10]. TILs recognize tumors and secrete pro-inflammatory cytokines with a role in the upregulation of PD-L1 expression in the TME. PD-L1 ligates PD-1 on TILs determining their downregulation [47]. The PD-L1 expression in TILs is interesting to keep in observation because PD-L1 was correlated with different degrees of lymphocytic infiltration described by the Clark system (X^2^ = 8.383, *p* = 0.020), according to references [48]. 

PD-L1 (+) expression shows a higher incidence in cases with superficial spreading melanoma, superficial spreading melanoma with vertical growth nodules, and a lower incidence rate in lentigo maligna melanoma and nodular melanomas, in conformity with references [49,50].

In the studies on melanomas, a high Breslow tumor thickness represents an indicator of poor prognosis [51,52,53]. PD-L1 expression was observed often in melanoma cases with more than 2−4 mm of Breslow tumor thickness [54,55]. According to the references, our study shows a significant correlation between the PD-L1 expression and Breslow tumor thickness in melanoma tumor cells (X^2^ = 9.933, *p* = 0.014).

Tumor-infiltrating lymphocytes (TILs) infiltrate tumor cells. The degree of infiltration is described by the TIL infiltrate’s extent and intensity. The most applied grading scheme for assessing the presence of TILs is the Clark system, as follows: 1; absent TILs infiltrate. 2; non-brisk TILs infiltrate: focal areas of lymphocytic infiltration in the tumor. 3; brisk TILs infiltrate: TILs infiltration of the entire tumor base or diffuse permeation of the tumor. Other systems for grading TILs infiltrates have been proposed based on their density and distribution but have not been validated [51,52,53]. TILs are an essential histopathological feature in diagnosing melanocytic tumors, composed of a series of cells, including T lymphocytes [56].

CD8+ cytotoxic and CD4+ T cells are important for the action of immune checkpoint inhibitors [38]. PD-L1 (+/-) expression patterns were observed on both tumor and immune cells. They were differentiated in the function of morphologic features, and tumors were considered to be PD-L1 positive when ≥5% of tumor cells presented membrane staining [57,58]. The presence of CD8+ TILs was scored as 0 (negative), weak (1+, positive cells focally at the edge of tumor or perivascular, 5% of tumor area), moderate (2+, infiltrating tumor by extending away from intratumoral vessels, 5−50% of tumor area), or intense (3+, broad infiltration by TILs, more than 50% of tumor area). Our study presented that PD-L1 expression was associated with CD8 TILs and CD4 TILs in the tumor immune microenvironment (X^2^ = 11.238, *p* = 0.003; X^2^ = 12.614, *p* = 0.001). These results are in accordance with multiple previous studies showing how PD-L1 positive expression represents an adaptive mechanism of expression induced by the host immune antitumor response [40,50,59,60]. 

In melanoma, PD-L1 is expressed by activated lymphocytes, presumably to decrease inflammation [61,62]. PD-L1 in stromal cells interacts with PD-1 on effector T cells, inhibiting their function. Moreover, in the tumor microenvironment, PD-L1 expression might be induced by interferon-gamma secretion via effector T cells [12]. Responding to immune checkpoint inhibitors may depend on active interaction between PD-1 on CD8 cells and PD-L1 in stromal cells [63].

PD-L1 plays an important role in the immune response regulation in the tumor microenvironment [64]. When PD-L1 binds to PD-1, it determines the T cell proliferation inhibition and its secretion of cytokines [65]. PD-1/PD-L1 blockade represents an essential strategy with therapeutic goals [66].

PD-L1 status was examined in TILs and MTCs cells, and cut-off values were calculated by the ROC curve method. Sensitivities of methods for PD-L1, CD8, and CD4 biomarkers, in patients with melanoma, were from 84.31% to 97.30%, and specificities of these analyzed biomarkers in melanoma tumor microenvironment cells were from 86.67% to 96.97%, offering the possibility to be biomarkers with predictive accuracy in melanomas. The area under the curve (AUC) for PD-L1, CD4, and CD8 biomarkers was from 0.941 to 0.975, which means a very good ability of the ROC test to discriminate the presence or absence of malign tumoral melanoma cells. To obtain more accurate measurements of PD-L1 expression, an IHC quantitative method associated with TILs characterization by CD4 and CD8 TILs expression was used, in conformity with other studies [67,68,69]. To verify antibody specificity, we used quality negative and positive controls for PD-L1 containing placental and tonsil tissue as previously described studies [57,67]. Many clinical assays develop predictive biomarkers by comparing the expression of PD-L1 in the tumor microenvironment cells in samples from patients treated with immune checkpoint inhibitors. Expressing these biomarkers in the tumor microenvironment predicts a better response to therapy [39].

One of the challenges to assessing the potential role of PD-L1 expression as a predictive biomarker was the variation in assays used across studies. Most studies used chromogenic IHC assays and obtained results of PD-L1 expression were interpreted by pathologists by reporting the percentage of tumor cells and immune cells demonstrating expression [14]. Other predictive biomarker studies involving PD-L1 expression used standard immunohistochemistry (IHC) as references [63,70,71,72].

We assessed the PD-L1, CD4, and CD8 lymphocytes infiltrations expression patterns, lymphovascular invasion, and perineural invasion in MTCs by multivariate analysis of Cox-proportional hazard regression. We observed that PDL-1 represents a favorable prognostic biomarker, being an independent predictor factor implied in the prognosis of the melanoma patients. 

The association between PD-L1 expression and prognosis in melanoma has been explored extensively in previous studies [32,33,40,57]. PD-L1 expression was an independent predictor of good prognosis in patients with melanomas. In some studies, PD-L1 expression was correlated with the treatment response, but other studies show it must be complemented by additional biomarkers for immune profiling of the CD8+ T cells [32,40,50,58,73].

Other studies demonstrated that a high expression rate of PD-L1 with a dense and diffuse T-cell inflammatory infiltrate is a characteristic feature of melanoma and represents a good prognostic marker for metastatic melanoma [74,75,76]. Adaptive PD-L1 expression in melanoma microenvironment pretreatment is associated with a good prognosis [57] and biomarker predictive of response to anti-PD-1/PD-L1 therapies in melanoma [38,70,72].

It is unclear if the PD-L1 negative status in advanced melanoma determinates phenotypic differences in immune responses or is an intrinsic tumor quality, and PD-L1–negative tumors are frequently associated with TILs. Melanoma with PD-L1–negative expression has a worse prognosis and responds less frequently to immune checkpoint therapies [11].

Lymphovascular invasion represents the presence of melanoma cells within the lamina of blood vessels or lymphatics, or both and is recorded as absent or present. Neurotropism represents the presence of melanoma cells close to the nerve sheaths. It is best identified at the periphery of the tumor and is associated with an increased local recurrence rate or local persistence. The density of lymphatic vessels within or surrounding growing tumors correlates with poor patient outcomes, being observed in many types of solid tumors [77,78]. Our data shows that lymphovascular and perineural invasion, both reliable morphologic prognostic features, did not correlate with PD-L1 expression and may not be considered an unfavorable prognostic marker. 

Many studies have investigated the impact of PD-L1 on the prognosis of solid tumors [79]. PD-L1 expression was an independent prognostic factor in renal cell carcinoma [80]. A high PD-L1 expression was a poor prognostic biomarker in patients with non-Hodgkin lymphoma [5,81].

Our results might have limitations such as a limited sampling, the lack of consensus method in the evaluation of PD-L1 expression (only quantitative in our study), not considering more categories of positivity depending on the amount and distribution of positive melanoma cells, and the fact that the immunohistochemical tests were performed on tissue fragments prior to any treatment. Other study limitations in the association between PD-L1 and melanoma prognosis are the heterogeneity, patient ethnicity, and treatment, which could influence patient survival. In further studies, we propose working with a larger cohort of melanoma patients to have meaningful results to validate the immuno-histochemical PD-L1 expression with roles in diagnosis and prognosis.

## 5. Conclusions

In this study, we used a melanoma patient cohort to characterize the PD-L1 expression in melanoma in the context of T-cell infiltrates because the PD-1/PD-L1 blockade represents the target in treating melanoma strategy. Most of the PD-L1 positive expressing tumors have a moderate score of CD4+ TILs and CD8+TILs (5-50% of tumor area) in tumoral melanoma environment cells. PD-L1 expression represents a predictive biomarker with very good accuracy to discriminate the presence or absence of malign tumoral melanoma cells. PD-L1 expression was an independent predictor of good prognosis in patients with melanomas.

## Figures and Tables

**Figure 1 jpm-13-00221-f001:**
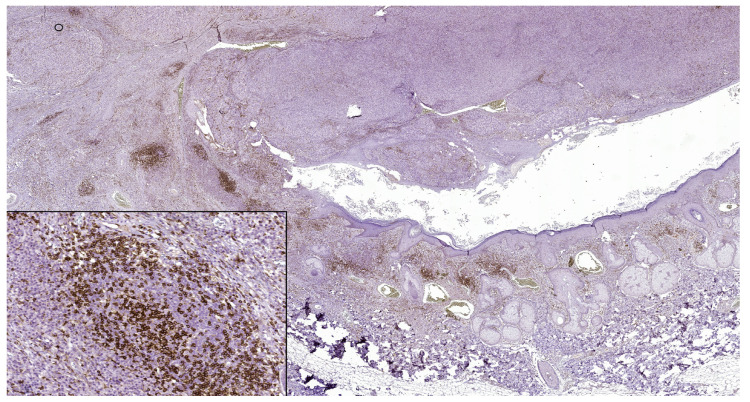
In nodular polypoid melanoma, CD4 intense membranous expression in intra and peritumoral lymphocytes (CD4 monoclonal antibody, clone EP204). In the lower left side detail, obx200.

**Figure 2 jpm-13-00221-f002:**
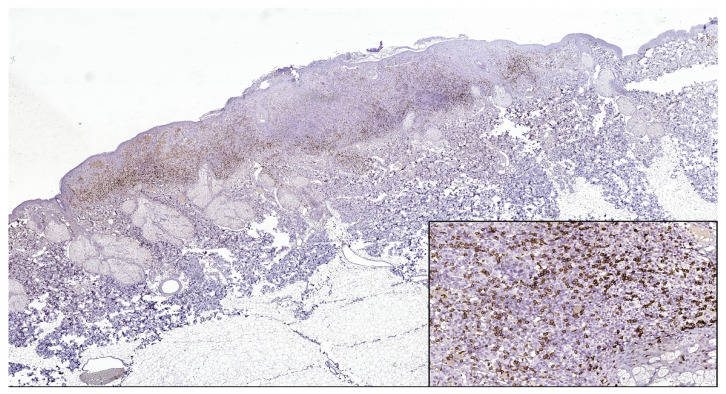
CD8 intense positive reaction in superficial spreading melanoma (CD8 Monoclonal Antibody, clone SP16, scanned image). In the lower right side, detail ob.x200.

**Figure 3 jpm-13-00221-f003:**
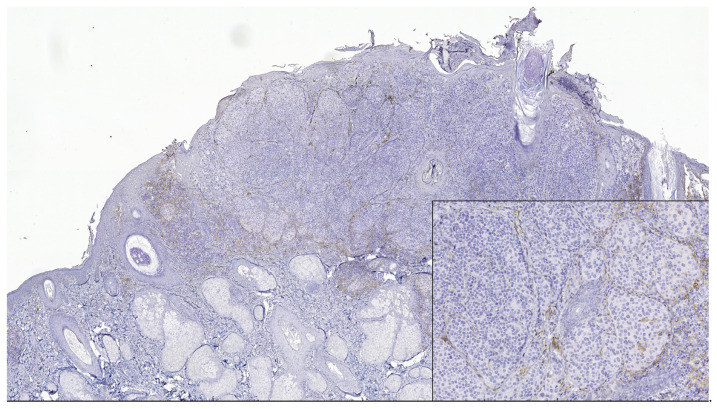
PD-L1 negative expression in tumor cells and TILs of nodular melanoma (PD-L1 Monoclonal Antibody, Clone CAL10). The brown staining is due to melanin pigment and melanophages. In the lower right side, detail ob.x100.

**Figure 4 jpm-13-00221-f004:**
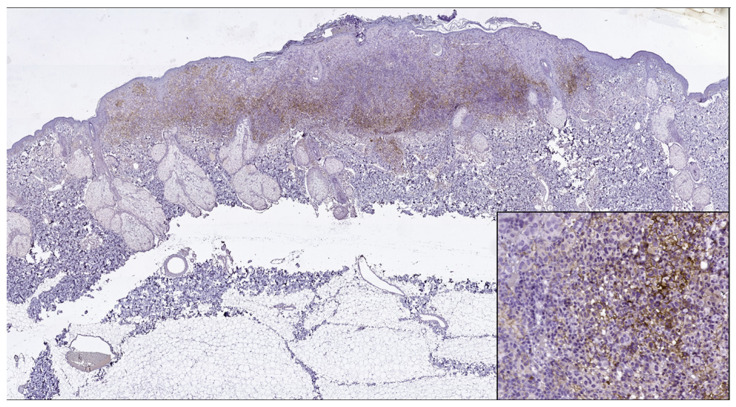
PD-L1 positive expression in melanoma cells of a superficial spreading melanoma (PD-L1 Monoclonal Antibody, Clone CAL10). Lower right side, detail, ob.x200.

**Figure 5 jpm-13-00221-f005:**
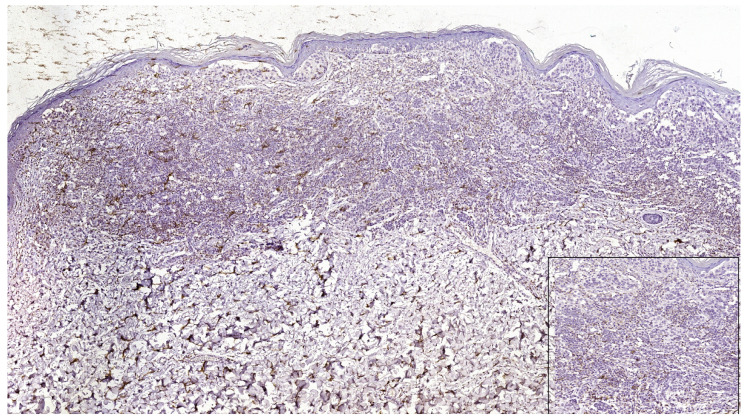
PD-L1 negative expression in tumor cells and focally positive in TILs in a case of superficial spreading melanoma (PD-L1 Monoclonal Antibody, Clone CAL10). Lower right side, detail, ob.x400.

**Figure 6 jpm-13-00221-f006:**
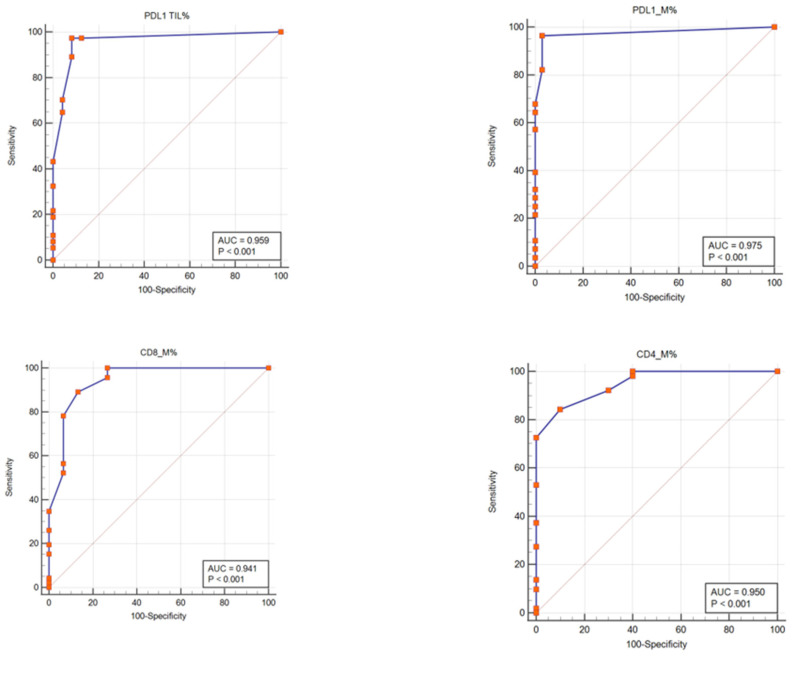
PD-L1, CD4, and CD8 tumor-infiltrating lymphocytes expressions patterns as potential biomarkers in melanoma diagnosis. Receiver operating curve (ROC) analyses were generated from 61 patients and had a value of the area under the curve (AUC) for PD-L1 (TILs: sensitivity: 97.30%, specificity: 91.67%; *p* < 0.001; M: sensitivity: 96.43%, specificity: 96.97%; *p* < 0.001), CD8 tumor-infiltrating lymphocytes (M: sensitivity: 89.13%, specificity: 86.67%; *p* < 0.001), and CD4 tumor-infiltrating lymphocytes (M: sensitivity: 84.31%, specificity: 90.00%; *p* < 0.001), respectively.

**Table 1 jpm-13-00221-t001:** Melanoma tumor microenvironment characterization in the function of PD-L1 (+/-) expression.

Nb.	Clinico- Pathological Aspects	PD-L1 Expression in TILs		X^2^ Score	*p*-Value	PD-L1 Expression in MTCs		X^2^ Score	*p*-Value
		Negative *n* (%)	Positive *n* (%)			Negative *n* (%)	Positive *n* (%)		
1.	**Age** <65 years >65 years	8 (33.33%) 16 (66.67%)	20 (54.05%) 17 (54.95%)	2.517	0.126	17 (51.52%) 16 (48.48%)	11 (39.29%) 17 (60.71%)	0.912	0.243
2.	**Gender** Males Females	9 (37.50%) 15 (62.50%)	22 (59.46%) 15 (40.54%)	2.809	0.120	15 (45.45%) 18 (54.55%)	16 (57.14%) 12 (42.86%)	0.828	0.444
3.	**Localization** Acral Upper body Cervico-cranial Limbs	3 (12.5%) 7 (29.17%) 9 (37.5% 5 (20.83%)	2 (5.41%) 19 (51.35%) 3 (8.11%) 13 (35.14%)	*9.603 **	*0.018*	3 (9.1%) 11 (33.33%) 7 (21.2%) 12 (36.36%)	2 (7.1%) 15 (53.62%) 5 (17.86%) 6 (21.43%)	2.791	0.443
4.	**Clark level of** **invasion** I II III IV V	6 (25%) 2 (8.33%) 4(16.67%) 11(45.83%) 1(4.17%)	3 (8.11%) 4 (10.81%) 6(16.22%) 21 (56.76%) 3 (8.11%)	3.584	0.494	7 (21.22%) 1 (3.03%) 3 (9.09%) 19 (57.58%) 3 (9.09%)	2(7.14%) 5 (17.86%) 7 (25.00%) 13 (46.43%) 1 (3.57%)	8.819	0.068
5.	**Breslow tumor thickness** <1 mm 1–2 mm 2–4 mm >4 mm	4 (16.67%) 0 (0%) 3 (12.50%) 17 (70.83%)	2 (5.41%) 3 (8.11%) 12 (32.43%) 20 (54.05%)	6.851	0.086	6 (18.18%) 0 (0%) 6 (18.18%) 21 (63.64%)	0 (0%) 3 (10.71%) 9 (32.14%) 16 (57.14%)	*9.933 **	*0.014*
6.	**Lymphovascular invasion** Present Absent	10 (41.67%) 14(52.14%)	24 (64.86%) 13(35.14%)	3.175	0.113	16 (48.48%) 17 (51.52%)	18 (64.29%) 10 (35.71%)	1.533	0.216
7.	**Perineural** **invasion** Present Absent	5 (20.83%) 19 (79.17%)	9 (14.75%) 33 (89.19)	1.163	0.298	5 (15.15%) 28 (84.85%)	4 (14.29%) 24 (85.71%)	0.009	0.926

PD-L1- Programmed cell death ligand-1; TILs-tumor-infiltrating lymphocytes; MTCs-melanoma tumor cells; * *p* < 0.05 represents statistically significant differences between cases made by Pearson Chi-Square with Cramer’s V test for the association between categorical variables, Mann-Whitney U Test and Kruskal-Wallis H Test for Continuous variables (X^2^ Score).

**Table 2 jpm-13-00221-t002:** Association between PD-L1 (+/-) expressions in tumor microenvironment cells and CD4 and CD8 tumor-infiltrating lymphocytes.

Nb.	Morphological Features	Infiltrating Immune Cells		X^2^ Score	*p*-Value	Melanoma Tumor Cells		X^2^ Score	*p*-Value
		PD-L1 (-) *n* (%)	PD-L1 (+) *n* (%)			PD-L1 (-) *n* (%)	PD-L1 (+) *n* (%)		
1.	**Morphologic type** LMM SSM SSM with VGN NM	2 (8.33%) 7 (29.17%) 6 (25%) 9 (37.50%)	1 (2.70%) 14 (37.84%) 14 (37.84%) 8 (21.62%)	3.305	0.332	3 (9.09%) 8 (24.24%) 13 (39.39%) 9 (27.27%)	0 (0%) 13 (46.43%) 7 (25%) 8 (28.57%)	5.678	0.144
2.	**TILs** **evaluation** Absent Brisk Non-brisk	8 (33.33%) 3 (12.50%) 13 (54.17%)	2 (5.41%) 8 (21.62%) 27 (72.97%)	*8.383 **	*0.020*	7 (21.21%) 4 (12.12%) 22 (66.67%)	3 (10.71%) 7 (25%) 18 (64.29%)	2.425	0.287
3.	**CD8 TIL’s** Negative <5% 5–50% >50%	9(37.50%) 1(4.17%) 14(58.33%) 0 (0.00%)	2 (5.41%) 1 (2.70%) 32 (69.57%) 2 (5.41%)	*11.238 ***	*0.003*	7 (21.21%) 0(0.00%) 25 (75.76%) 1 (50.00%)	4 (14.29%) 2 (7.14%) 21 (75.00%) 1 (50.00%)	2.775	0.514
4.	**CD4 TIL’s** Negative <5% 5–50% >50	6 (25.00%) 1 (4.17%) 17 (70.83%) 0 (0.00%)	0 (0.00%) 0 (0.00%) 36 (97.30%) 1 (2.70%)	*12.614 ***	*0.001*	3 (9.09%) 0 (0.00%) 29 (87.88%) 1 (3.03%)	3 (10.71%) 1 (3.57%) 24 (85.71%) 0 (0.00%)	2.076	0.906

PD-L1- Programmed cell death ligand-1; TILs- tumor-infiltrating lymphocytes; LMM- lentigo maligna melanoma; SSM- Superficial spreading melanoma; SSM-VGN- superficial spreading melanoma with vertical growth nodule; NM- nodular melanoma; * *p* < 0.05 and ** *p* ≤ 0.001 represent statistically significant differences between cases made by Pearson Chi-Square with Cramer’s V test for the association between categorical variables, Mann-Whitney U Test and Kruskal-Wallis H Test for Continuous variables (X^2^ Score).

**Table 3 jpm-13-00221-t003:** Receiver operating characteristic (ROC) analysis of PD-L1, CD4, and CD8 tumor-infiltrating lymphocyte biomarkers in melanoma patients.

Nb.	Biomarkers	AUC	95% CI *	*p*-Value	Youden J Index	Cut-Off Value	Sensitivity	Specificity
1.	**PD-L1 TILs**	0.959	0.875 to 0.993	<0.001	0.889	>3	97.30	91.67
2.	**PD-L1 M**	0.975	0.898 to 0.998	<0.001	0.934	>0	96.43	96.97
3.	**CD4 M**	0.950	0.862 to 0.989	<0.001	0.743	>15	84.31	90.00
4.	**CD8 M**	0.941	0.850 to 0.985	<0.001	0.758	>10	89.13	86.67

UC = area under the curve; ROC = receiver operating characteristic; * Confidence interval; Youden J = sensitivity + specificity – 100; PD-L1- Programmed cell death ligand-1; TILs- tumor-infiltrating lymphocytes; M- melanoma tumor cells.

**Table 4 jpm-13-00221-t004:** Logistic regression of potential prognostic values of PD-L1, CD4, and CD8 TILs in melanoma tumor microenvironment cells.

Nb.	Clinical Variables	Multivariate Analysis		
		Melanoma Tumor Microenvironment Cells		
		Hazard Ratio	*p*-Value	95% CI *
1.	Lymphovascular invasion	3.049	0.528	0.095−97.914
2.	Perineural invasion	0.026 *	** *0.023* **	0.001−0.607
3.	PD-L1	2.700	0.455	0.198−36.683
4.	CD4 TILs	11.838 *	** *0.023* **	1.386−101.108
5.	CD8 TILs	7.748 *	** *0.018* **	1.408−42.622

CI^∗^- Confidence interval; * *p* < 0.05 represents statistically significant differences between variables made by Cox Logistic & Snell regression (Cox & Snell R^2^). PDL-1 in MTC-R^2^ = 0.23, *p* = 0.43; CD8 TILs in MTC – R^2^ = 0.49, *p* < 0.0001; CD4 TILs in MTC - R^2^ = 0.38, *p* = 0.002.

## Data Availability

Data are contained within the article.

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
