# Peer review of "PD-L1, CD4+, and CD8+ Tumor-Infiltrating Lymphocytes (TILs) Expression Profiles in Melanoma Tumor Microenvironment Cells"

_jpm, 2023, doi:10.3390/jpm13020221_

Round 1

Reviewer 1 Report

The manuscript by Bogdan Marian Caraban et al. describes studies on PD-L1, CD4+, and CD8+ tumor-infiltrating lymphocytes expression profiles in melanoma tumor microenvironment cells. It was found that most of the PD-L1 positive expressing tumors have a moderate score of CD4+TILs, and CD8+ TILs in tumoral melanoma environment cells. The PD-L1 expression in TILs was correlated with different degrees of lymphocytic infiltration. PD-L1 expression was observed often in melanoma cases. It indicated that PD-L1 expression represents a predictive biomarker with very good accuracy to discriminate the presence or absence of malign tumoral melanoma cells. PD-L1 expression was an independent predictor of good prognosis in patients with melanomas. This manuscript is well written and presents interesting new findings on PD-1/PD-L1 immune inhibitory signaling pathways in melanoma cases, which might be associated with the potential diagnosis and prognosis.

Some minor issues that require the authors to address:

1, PD-L1 is one of the two ligands of receptor PD-1 induced on effector T lymphocytes in response to inflammatory signals. Engagement and interaction of PD-1/PD-L1 play an inhibitory role in T cell-mediated immune response. In this manuscript, it is necessary to investigate the PD-1 receptor expression profiles in the CD4+ and CD8+ T lymphocytes which is important to confirm the conclusions. However, the authors did not show the data in the manuscript.

2, Besides, it is also meaningful to detect expression of the related cytokines produced by CD4+ and CD8+ T lymphocytes, such as IFN-gamma, TNF-alpha, IL-2, IL-6, IL-10, in melanoma. It is suggested that the authors should supplement the data in the manuscript.  

3, In the part of Discussion, it is also need to be revised for better clarity of description on the regulatory roles of the PD-1/PD-L1 inhibitory signaling in tumor immune evasion, which is favorable to improve the scientific quality of the manuscript.

Author Response

Journal of Personalized Medicine, 21.01.2023

                                                                   Dear Reviewer,

Thank you, for your kind comments and recommendations. The corrections have been implemented in the manuscript.

Reviewer # 1-Open Review

 ( ) I would not like to sign my review report

(x) I would like to sign my review report

English language and style

( ) English very difficult to understand/incomprehensible

( ) Extensive editing of English language and style required

( ) Moderate English changes required

(x) English language and style are fine/minor spell check required

( ) I don't feel qualified to judge about the English language and style

Yes           Can be improved            Must be improved          Not applicable

Does the introduction provide sufficient background and include all relevant references?

(x)                       ( )                                              ( )                                  ( )

Are all the cited references relevant to the research?

(x)                        ( )                                             ( )                                  ( )

Is the research design appropriate?

( )                         (x)                                             ( )                                 ( )

Are the methods adequately described?

(x)                         ( )                                              ( )                                ( )

Are the results clearly presented?

( )                          (x)                                             ( )                                 ( )

Are the conclusions supported by the results?

( )                          (x)                                              ( )                                ( )

Comments and Suggestions for Authors

The manuscript by Bogdan Marian Caraban et al. describes studies on PD-L1, CD4+, and CD8+ tumor-infiltrating lymphocytes expression profiles in melanoma tumor microenvironment cells. It was found that most of the PD-L1 positive expressing tumors have a moderate score of CD4+TILs, and CD8+ TILs in tumoral melanoma environment cells. The PD-L1 expression in TILs was correlated with different degrees of lymphocytic infiltration. PD-L1 expression was observed often in melanoma cases. It indicated that PD-L1 expression represents a predictive biomarker with very good accuracy to discriminate the presence or absence of malign tumoral melanoma cells. PD-L1 expression was an independent predictor of good prognosis in patients with melanomas. This manuscript is well written and presents interesting new findings on PD-1/PD-L1 immune inhibitory signaling pathways in melanoma cases, which might be associated with the potential diagnosis and prognosis.

Some minor issues that require the authors to address:

 PD-L1 is one of the two ligands of receptor PD-1 induced on effector T lymphocytes in response to inflammatory signals. Engagement and interaction of PD-1/PD-L1 play an inhibitory role in T cell-mediated immune response. In this manuscript, it is necessary to investigate the PD-1 receptor expression profiles in the CD4+ and CD8+ T lymphocytes which is important to confirm the conclusions. However, the authors did not show the data in the manuscript.

Response:

As your suggestions, in table 2 of this manuscript were presented the PD-1 (+/-) receptor expression profiles associated with CD4+ and CD8+TILs.

“PD-L1 (+) patterns associated with moderate to severe of CD8+ and CD4+ lymphocytes intensities (CD8+TILs: 69.57%; MTCs: 75.00%; CD4+TILs: 97.30%; MTCs: 85.71%) compared to mild (CD8+TILs: 2.79%; MTCs: 7.14%, CD4+TILs: 0.00%; MTCs: 3.57%, p<0.05) were observed in majority of morphological types of melanomas (table 2). Melanomas negative for PD-L1 show significant degree of staining (2+) by CD4 and CD8 infiltrating lymphocytes (CD8+TILs: 58.53%; MTCs: 75.76%, CD4+TILs: 70.83%; MTCs: 87.88%, p<0.05).”

  1. Besides, it is also meaningful to detect expression of the related cytokines produced by CD4+ and CD8+ T lymphocytes, such as IFN-gamma, TNF-alpha, IL-2, IL-6, IL-10, in melanoma. It is suggested that the authors should supplement the data in the manuscript.

Response:

Your idea to detect the expressions of the related cytokines produced by CD4+ and CD8+ T lymphocytes, such as IFN-gamma, TNF-alpha, IL-2, IL-6, IL-10, in melanoma cases is interesting, but for this manuscript we didn’t study these related cytokines, but in the future articles we shall use your suggestion.

  1. In the part of Discussion, it is also need to be revised for better clarity of description on the regulatory roles of the PD-1/PD-L1 inhibitory signaling in tumor immune evasion, which is favorable to improve the scientific quality of the manuscript.

Response:

As you suggested, we clarified the description about the regulatory roles of the PD-1/PD-L1 inhibitory signaling in tumor immune evasion.

“Melanomas may express PD-L1 against antitumor immune effector cells, facilitating immune evasion even if the B7-H1 costimulatory molecule is present on the surface of tumor cells. This regulates the cellular and humoral immune responses through the PD-1 receptor on activated T and B cells. Also, it was observed that in vitro, the B7-H1 tumor cell lines might increase the apoptosis of antigen-specific human T-cells, which means the apoptotic effects of B7-H1 are mediated by one or more receptors other than PD-1.”

Yours faithfully,

Ph.D. Biologist Matei Elena

 [email protected]

Reviewer 2 Report

In this study, the authors characterized the expression of PD-L1 and investigated its association with tumor-infiltrating lymphocytes in treatment-naïve melanoma specimens. Via immunohistochemical staining, ROC and multivariate regression analyses, the authors assessed the accuracy of PD-L1, CD4 and CD8 as potential biomarkers for melanoma diagnoses, and highlighted the role of PD-L1 as a prognostic biomarker. This is a well-written manuscript, with interesting findings of importance to the field of immunology and biomarkers for melanoma.

I have a few queries below:

1.      How was the PD-L1 scored? Was it scored specifically on lymphocytes (i.e., CD4 and CD8 T-cells expressing PD-L1 specifically), or on all immune cells e.g., macrophages? Could the authors please include these details in the methods section?

2.      In Table 2, if TILs are absent, could the authors please explain why there are 2 samples with PD-L1+ infiltrating immune cells?

3.      Table 3 and Figure 6 are not clear to me. Why do they refer to CD4 and CD8 ‘M’? Should this be TILs instead? 

1

Author Response

Journal of Personalized Medicine, 21.01.2023

                                                                   Dear Reviewer,

Thank you, for your kind comments and recommendations. The corrections have been implemented in the manuscript.

Reviewer # 2-Open Review

 (x) I would not like to sign my review report

( ) I would like to sign my review report

English language and style

( ) English very difficult to understand/incomprehensible

( ) Extensive editing of English language and style required

( ) Moderate English changes required

(x) English language and style are fine/minor spell check required

( ) I don't feel qualified to judge about the English language and style

Yes           Can be improved            Must be improved          Not applicable

Does the introduction provide sufficient background and include all relevant references?

(x)                         ( )                                         ( )                                     ( )

Are all the cited references relevant to the research?

(x)             ( )           ( )           ( )

Is the research design appropriate?

(x)             ( )           ( )           ( )

Are the methods adequately described?

( )              (x)          ( )           ( )

Are the results clearly presented?

( )              (x)          ( )           ( )

Are the conclusions supported by the results?

( )              (x)          ( )           ( )

Comments and Suggestions for Authors

In this study, the authors characterized the expression of PD-L1 and investigated its association with tumor-infiltrating lymphocytes in treatment-naïve melanoma specimens. Via immunohistochemical staining, ROC and multivariate regression analyses, the authors assessed the accuracy of PD-L1, CD4 and CD8 as potential biomarkers for melanoma diagnoses, and highlighted the role of PD-L1 as a prognostic biomarker. This is a well-written manuscript, with interesting findings of importance to the field of immunology and biomarkers for melanoma.

I have a few queries below:

 How was the PD-L1 scored? Was it scored specifically on lymphocytes (i.e., CD4 and CD8 T-cells expressing PD-L1 specifically), or on all immune cells e.g., macrophages? Could the authors please include these details in the methods section?

Response:

As you suggested, please see the description of the PD-L1 expression on cells included in the methods section.

“We assessed the PD-L1 expression in melanoma cells and TIL’s (tumor-infiltrating lymphocytes), and CD4 and CD8 TIL’s expressions in melanoma tumor microenvironment cells.”

 In Table 2, if TILs are absent, could the authors please explain why there are 2 samples with PD-L1+ infiltrating immune cells?

Response:

The expression of the PD-L1 may be positive even if TILs are absent because all assessment is useful to establish the accuracy of the method.

  1. Table 3 and Figure 6 are not clear to me. Why do they refer to CD4 and CD8 ‘M’? Should this be TILs instead?

Response:

Table 3 and figure 6 present the ROC analysis of PD-L1, CD4, and CD8 tumor-infiltrating lymphocyte biomarkers in melanoma patients, necessary to establish the accuracy of the method, implied in the diagnostic.

As we explained in the methods section the PD-L1 expressions were assessed in melanoma cells (M) and TIL’s (tumor-infiltrating lymphocytes) and this is the reason why appear in the table 3 and figure 6 such as PD-L1 TILs and PD-L1 M.

Also, CD4 and CD8 TIL’s expressions were assessed only in melanoma tumor microenvironment cells (M) and this is the reason why appear in the table 3 and figure 6 such as CD4 M and CD8 M.

Yours faithfully,

Ph.D. Biologist Matei Elena

 [email protected]
